# FDS-Based Study of the Fire Performance of Huizhou Fire Seal Walls in Traditional Residential Buildings in Southern China

**Yunfa Wu [1,2], Bin Hua [1], Sarula Chen [1,*] and Jimo Yang [1]**

[1] School of Architecture and Planning, Anhui Jianzhu University, Hefei 230601, China;
wuyf@ahjzu.edu.cn (Y.W.); hb1@stu.ahjzu.edu.cn (B.H.); yangjimo@ahjzu.edu.cn (J.Y.)

[2] State Key Laboratory of Fire Science, University of Science and Technology of China, Hefei 230026, China

[*] Correspondence: sarul@tju.edu.cn

**Abstract:** In the history of human civilization, traditional villages and buildings have been significantly threatened by fire. As an essential part of Huizhou traditional architecture, fire seal walls play a crucial role in preserving Huizhou architecture by effectively blocking the spread of fire. However, with economic and social development, the Huizhou fire seal wall's original fire prevention function has been unable to meet the needs of modern fire protection. This study aims to explore the fire performance of different types of Huizhou fire seal walls to provide a reference guide for future fire protection, optimization, and transformation of traditional buildings. In this paper, 3D models of traditional buildings with fire seal walls were built with FDS, and the performance of the different kinds of fire seal walls was simulated under the influence of wind speeds, building spacing, and the height of the vertical ridge of the fire seal wall. The results showed that, under the same conditions, a fire seal wall with a single eave is superior to fire seal walls with quintuple eaves in terms of performance, and fire seal walls with quintuple eaves are superior to fire seal walls with triple eaves in the middle and late stages of a fire. In addition, wind speeds, building spacing, and the height of the vertical ridge have different effects on the fire performance of seal walls. Lower wind speeds can reduce the fire performance of fire seal walls, and no wind and higher wind speeds have no significant effect on the fire performance of fire seal walls, while increasing the height of the vertical ridge and fire spacings appropriately can improve the fire performance of fire seal walls. This study provides a reference guide for future fire protection, optimization, and transformation of Huizhou fire seal walls, which can help improve the fire safety of traditional buildings.

**Keywords:** fire performance; fire spread; Huizhou fire seal walls; traditional buildings; FDS

## 1. Introduction

### 1.1. Fire Prevention in Traditional Chinese Buildings

China has a long history and rich cultural heritage as one of the four ancient civilizations. Traditional villages and buildings have been widely distributed throughout the country as cultural heritage objects with significant regional and ethnic characteristics [1], which not only have historical research value [2] but also contribute to the inheritance of traditional culture [3], improvement in infrastructure [4], and development of agriculture and tourism [5,6]. As stated by the Chinese Ministry of Housing and Urban-Rural Development, 8171 villages were included in the List of Traditional Villages of China by the end of 2022 [7], which rendered China the most significant cultural heritage conservation country worldwide [8]. However, with recent economic and social advancements, China has experienced frequent fires, which pose a severe threat to the conservation of traditional villages and buildings, and many traditional villages and buildings are destroyed by fire every year [9]. For example, a fire broke out in the ancient city of Dukesong in Yunnan Province, China, in 2014, covering an area of nearly one hundred and one acres, causing the destruction of 242 old buildings and direct property damage amounting to 89.83 million yuan. A fire

broke out in an ancient town in Nanjiang County, Sichuan Province, China, in 2016, causing 26 people to be affected and property damage totaling more than 3 million. This entails not only historical and cultural heritage loss but also significant damage to the local natural environment and human landscape. There are four main reasons for fires. First, traditional Chinese buildings are mainly constructed of wooden structures, which have been eroded by wind and rain for thousands of years and exhibit a low water content; thus, they can easily ignite once they encounter open fires [10]. Second, traditional Chinese villages and buildings are mainly located in remote rural areas, which usually have treacherous terrain and roads, making it difficult for fire engines to enter and extinguish fires [11]. Third, traditional Chinese villages are densely distributed with small building spacing, which promotes the occurrence of mass fires. Fourth, firefighting facilities in rural areas must be improved, and public awareness of firefighting must be enhanced. It is difficult to extinguish fires the first time they occur. Therefore, it is still a significant problem that fire threatens the conservation of traditional buildings and villages [12]. The study of fire protection in traditional villages and buildings is important within this context.

### 1.2. Fire Research in Traditional Buildings

Fire protection research for traditional villages and buildings is currently focused on the following four aspects: First, fire risk assessment research [13,14] mainly identifies potential or existing fire risks of target objects using field research and provides assessment levels using hierarchical analysis, fuzzy analysis, and accident tree analysis [15], and corresponding fire protection optimization strategies are finally proposed in a targeted manner. Secondly, numerical simulation research of fires [16–18] mainly adopts the FDS software (https://pages.nist.gov/fds-smv/downloads.html) simulation method to establish a traditional village or building numerical model so that it can be employed to study the changes in flame plume, smoke spread, and heat radiation conditions during a fire. This method dramatically reduces the cost of fire research and provides high experimental safety and economy. Thirdly, physical experimental combustion simulation research [19–21] entails the construction of a scaled or full-size model to study the combustion characteristics of wood and fire spread patterns between buildings. This measurement method is more accurate than numerical simulation, but the economic cost and danger are significantly higher. Fourthly, traditional fire protection technology research [22], which favors the theoretical level, focuses on compiling and summarizing ancient villages and building fire protection experiences using historical literature and field research and providing corresponding explanations. For example, Chinese patios and fire alleys are ancient fire prevention techniques that have saved countless houses from disasters for hundreds of years. From the above, current research on fire prevention in traditional villages and buildings is a focus of research from theoretical to practical levels, and it is conventional relative to present technology levels.

However, from microscopic combustion simulation to macroscopic fire risk assessment levels, research on traditional fire prevention technology remains at the theoretical research level, with few scholars verifying conventional fire protection measures using modern technology and proposing updated alternatives. Moreover, with the development of contemporary society, traditional fire protection systems are gradually disintegrating and disappearing, and the original fire protection measures can no longer fully meet the requirements of village or building protection; thus, it is essential to conduct a modern fire protection transformation of traditional fire protection technology.

### 1.3. Huizhou Fire Seal Walls

Huizhou fire seal walls are called Matou walls because the top base of the wall is shaped to resemble a horse's head. Huizhou fire seal walls are usually made of brick and stone; they are built around wooden buildings and are mostly constructed with hollow bucket walls because they do not bear weight [23]. Therefore, the bricks in Huizhou are large and thin. The fire seal wall is built freely, roughly following the rule that the lower

solid fence should be covered with air and the empty wall should not be connected; the wall is usually painted with white ash, and the eaves are covered with very small green tiles [24], forming a powder wall with tiles, which is very decorative. There are three main parts:

1. Wall;
2. Plucked leaves, pallets, and pallet heads, for which the primary function is to prevent the wall from being directly impacted and soaked by rain;
3. The ridge of the Matou wall is the closing part of the sealed firewall.

According to the number of eaves on the wall above the roof, they can be classified as fire seal walls with a single eave, fire seal walls with triple eaves, and fire seal walls with quintuple eaves [25] (Table 1). Huizhou fire seal walls, primarily located in the Huizhou region of China, are a traditional Chinese building fire protection technique that can effectively prevent flames from spreading to adjacent buildings in the event of a fire, thus reducing the danger. Most existing Huizhou fire seal walls originated during the Ming and Qing Dynasties, when He San, the governor of Huizhou, ordered residents to adjust the height of walls for fire protection, after which fire seal walls were widely used for fire protection in the Huizhou area. For hundreds of years, they have been used as essential fire protection technology to protect Huizhou's traditional villages and buildings from the threat of fire and safeguard the lives and property of countless people.

**Table 1.** Classification of Huizhou fire seal walls.

| Type | Number of Eaves | Example Photos | | | Schematic |
|------|-----------------|----------------|---|---|-----------|
| fire seal wall with a single eave | 1 | | | | |
| fire seal wall with triple eaves | 3 | | | | |
| fire seal wall with quintuple eaves | 5 | | | | |

However, with the development of modern society, the fire protection function of the Huizhou fire seal wall has gradually weakened, and the original fire protection system and architectural style have been seriously damaged; thus, traditional Huizhou fire seal walls can no longer meet modern fire protection demands.

From the above, the fire prevention situation of traditional Chinese buildings and villages is becoming increasingly serious, and traditional fire prevention technology is gradually dying out. As a traditional fire prevention technology, Huizhou fire seal walls have very important fire prevention research value, but there is little research in related fields. Therefore, it is necessary to further strengthen the role of traditional fire protection technology for traditional village and building conservation. Inspired by this aspect, this study established the following research steps (Figure 1): First, we visited traditional villages in Huizhou, conducted field research on the Huizhou fire seal wall of the traditional Chinese fire protection system, and selected a typical dwelling for mapping. Secondly, the Huizhou fire seal wall was numerically simulated using FDS software (https://pages.nist.gov/fds-smv/downloads.html). The purpose was to compare fire resistance differences

and the weak points of fire seal walls with single eaves, triple eaves, and quintuple eaves under the same conditions and analyze the effects of wind speeds, building spacing, and vertical ridge height on the fire resistance of Huizhou fire seal walls under different fire scenarios. Finally, corresponding Huizhou fire seal wall renovation strategies were proposed, which provide a practical reference for the future fire protection renovation of the same type of fire seal walls, and this fundamentally contributes to the protection of traditional villages and buildings in Huizhou.

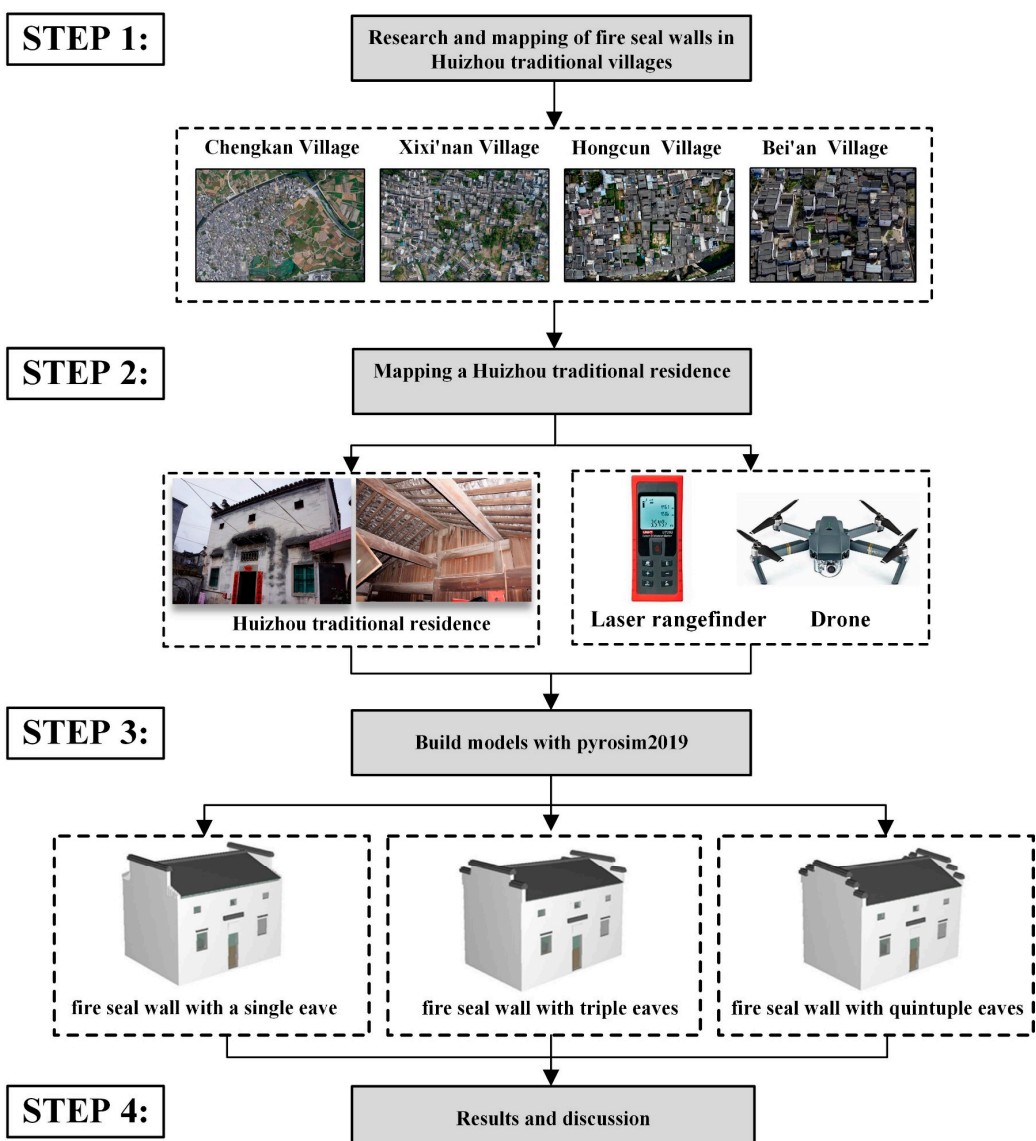

**Figure 1.** Schematic diagram of the workflow of this study.

## 2. Materials and Methods

### 2.1. Study Area and Data Sources

2.1.1. Study Area

The selected study area is in Huangshan City, Anhui Province, China (Figure 2a), which is in the southern part of Anhui Province, East China, with a total area of approximately 9800 km², between 117°02′–118°55′ E and 29°24′–30°24′ N. The resident population of Huangshan city is 1,332,000 people, of which the urban population accounts for 59.25% of the total population and the rural population accounts for 40.75% of the total population. Its subtropical monsoon climate results in four distinct seasons, sufficient rainfall, and much heat. According to a typical Chinese meteorological year query, the average wind

speed is 7 m/s at maximum and 1.6 m/s at minimum throughout the year. The average monthly maximum value of the dry bulb temperature is 30 °C, and the average monthly minimum value is 10 °C. In addition, it is also the birthplace of Hui culture, for example, Huizhou architecture, Huizhou four sculptures, and Huizhou opera. Among them, Hui-style architecture [26], as historical and cultural heritage, is distributed in large numbers among various ancient villages in Huangshan city, such as Xixi'nan Village, Hongcun Village, and Chengkan Village, all World Heritage Sites [27]. Therefore, this study adopted the protection of traditional villages and buildings in Huizhou as the starting point of the current severe form of traditional village and building fire protection and used FDS to numerically simulate the Huizhou fire seal wall, which can provide important reference values for the optimization of traditional fire protection technology in Huizhou and the protection of historical and cultural heritage.

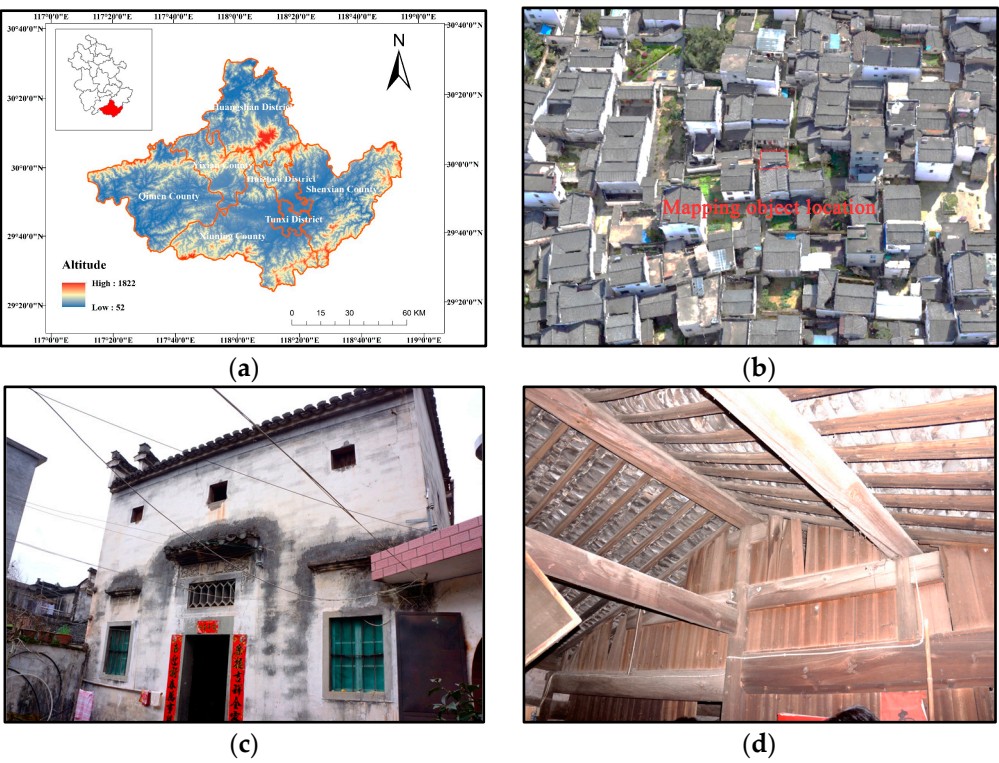

**Figure 2.** Study area and mapping objects. (**a**) Location map of Huangshan, China; (**b**) General plan of Bei'an village; (**c**) Exterior view of the mapping object; (**d**) Interior view of the mapping object.

2.1.2. Data Sources

The study conducted research and mapping in Huangshan City. The measuring tools were a laser rangefinder and drone, and CAD computer software (https://www.autodesk.com.hk/solutions/cad-software) was adopted to plot the measurement results. Several traditional Huizhou villages, such as Chengkan Village, Xixi'nan Village, Hongcun Village, and Bei'an Village, were visited to investigate different Huizhou fire seal walls during the 2021–2022 period. In addition, traditional residential buildings were better preserved in Bei'an Village. Therefore, to facilitate the numerical simulation of fire seal walls, a typical traditional Huizhou residence with fire seal walls (Figure 2b–d) in Bei'an Village was selected as a study object for mapping. The mapping's results show a building with fire seal walls with triple eaves on the eastern and western sides; the walls are painted white lime, the rugged hill-shaped roof is covered with black tiles, and the interior structure is a Chinese beam-bearing wooden frame. With a total height of 2 floors (Figure 3), the building area is 75 m$^2$, and the floor space is 37.5 m$^2$. Constructed during the Chinese Qing Dynasty (1664–1911), the building has essential historical research value.

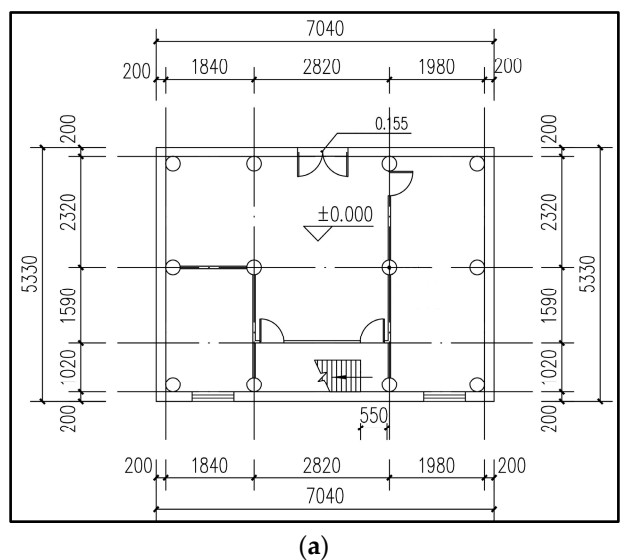 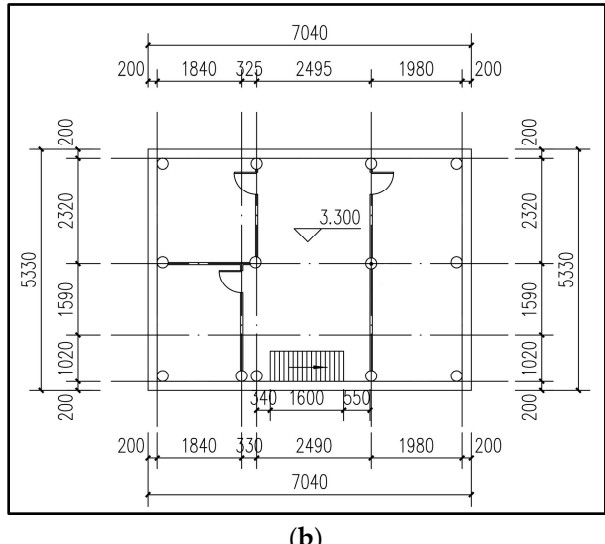

(**a**)                                                 (**b**)

**Figure 3.** Plan of the mapping object. (**a**) First-floor plan; (**b**) second-floor plan.

*2.2. Experimental Design and Boundary Condition*

2.2.1. Experimental Design

This study refers to the above mapping data and uses Pyrosim 2019 to build the model. A single model of a fire seal wall with triple eaves with a length of 7040 mm, a width of 5330 mm, and a height of 6900 mm, including a total of 2 floors, is established along the positive direction of the *X*-axis, and the front elevation is oriented along the negative direction of the *Y*-axis. Among them, the exterior wall material is masonry, and the interior columns, beams, squares, interior partition walls, and floor slabs are set to yellow pine (Table 2). The thickness of the exterior walls is set to 200 mm, the cross-sectional size of the interior columns is set to 200 mm × 200 mm, the cross-sectional size of the beams and square is set to 150 mm × 150 mm, the thickness of the partition wall is set to 50 mm, and the thickness of the floor slab is set to 100 mm. The roof material is developed into green tiles; each piece is 150 mm × 200 mm in size. The sizes of the exterior windows are set to 870 mm × 1050 mm and 400 mm × 320 mm, for a total of 2 types. After this model is established, monolithic models of fire seal walls with single eaves and fire seal walls with quintuple eaves are similarly established, according to the mapping data of the Huizhou fire seal walls researched. The parameters of the other parts are consistent with those of the established model of the fire seal wall with triple eaves, except for the eastern and western side walls.

**Table 2.** Material parameters.

| Name of Material | Specific Heat Capacity (kJ/kg·K) | Density (kg/m³) | Conductivity (W/m·K) | Calorific Value (kJ/kg) |
|---|---|---|---|---|
| Yellow pine | 2.3 | 570 | 0.2 | 18,000 |
| Fire seal wall | 1.05 | 1700 | 1.89 | |
| Tile | 1.24 | 1200 | 0. 43 | |

The fire simulation scenarios in this paper are as follows: fire seal walls with single eaves, fire seal walls with triple eaves, and fire seal walls with quintuple leaves are selected as fire simulation objects based on the above study. Twenty-seven fire simulation scenarios (Figure 4) were constructed by selecting fire-spreading influencing factors from three different levels. At the meteorological level, wind speed was chosen as the most critical factor affecting fire spread. There were three wind speeds (0 m/s, 1.6 m/s, and 7 m/s), representing no wind, the lowest annual average wind speed, and the highest average yearly wind speed in Huizhou, respectively, and these scenarios were recorded as W1–W9.

At the street level, the building spacing was selected as the most critical factor affecting fire spread, in which there were three building spacings (0.8 m, 1.6 m, and 2.4 m), and these scenarios were recorded as F1–F9. At the building level, the height of the vertical ridge of the fire seal walls was selected as the most influential factor of fire spread, among which there were three vertical ridge heights (0 m, 0.5 m, and 1 m), representing the standard heights of the vertical ridges of Huizhou fire seal walls, and these scenarios were recorded as H1–H9. The other factors are set as follows: the environmental temperature is 20 °C, the wind direction is west, and the building location is parallel.

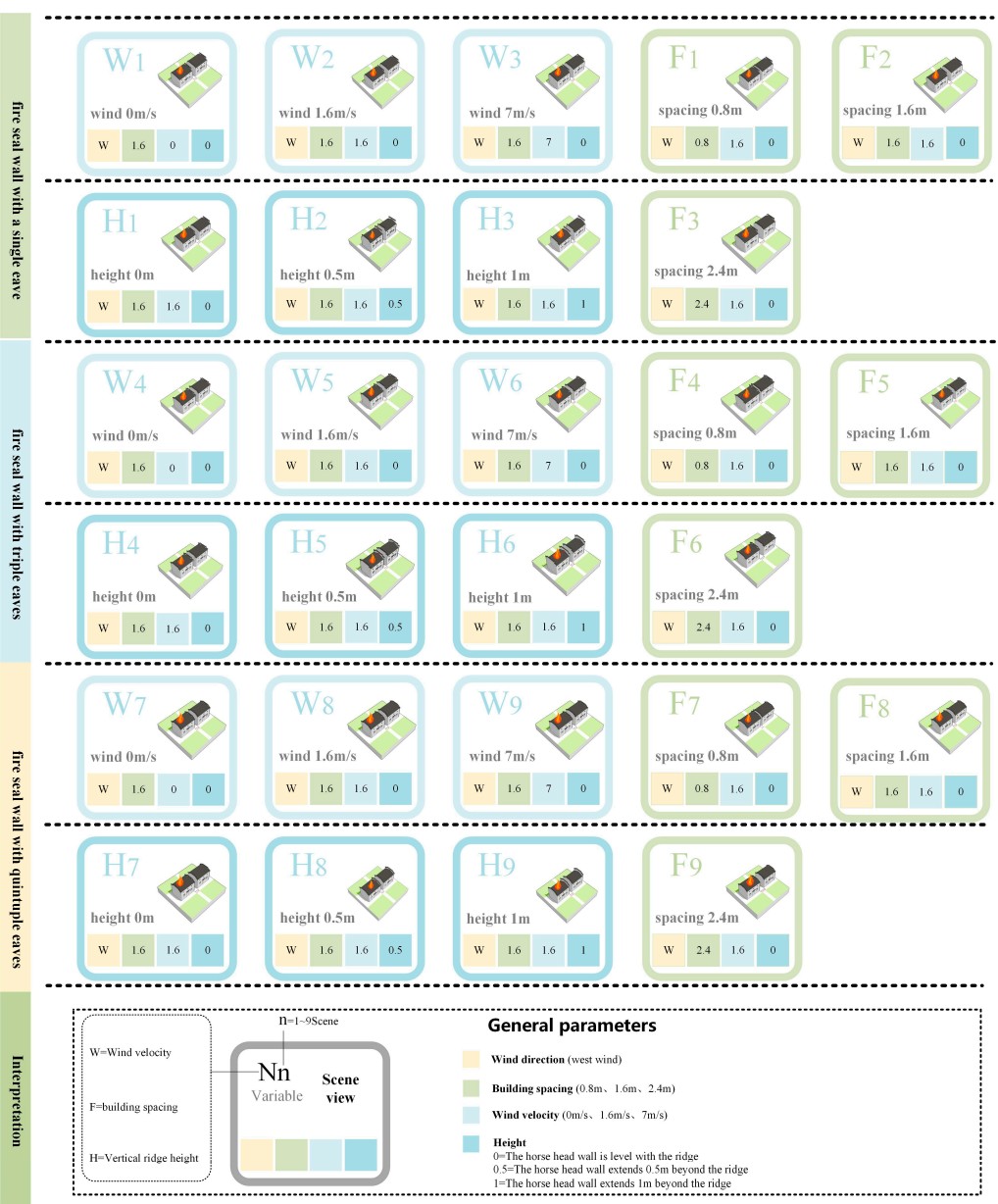

**Figure 4.** Fire simulation scenario design.

### 2.2.2. Equipment Arrangement

Temperature slices (Figure 5) were arranged at 0.8 m, 1.6 m, and 2.4 m from the right side of the fire seal wall under the 27 simulated scenarios and kept parallel to the fire seal wall surface. Temperature detectors were arranged in adjacent temperature slices, and thermocouples were placed vertically at 0.5 m intervals in the middle and on both sides, with 33 in total. The detectors at 0.8 m intervals on the fire seal wall with a single eave were recorded as N1–N33; the highest point in the middle is N13; the highest point on the left is

N33; and the highest point on the right is N23 (Table 3). The detectors at 0.8 m intervals on the fire seal wall with triple eaves are recorded as T1–T33, with the highest point in the middle denoted as T13, the highest point on the left denoted as T33, and the highest point on the right denoted as T23. The detectors at 0.8-m intervals on the fire seal wall with quintuple eaves are marked as F1–F33, with the highest point in the middle marked as F13, the highest point on the left marked as F33, and the highest point on the right marked as F23.

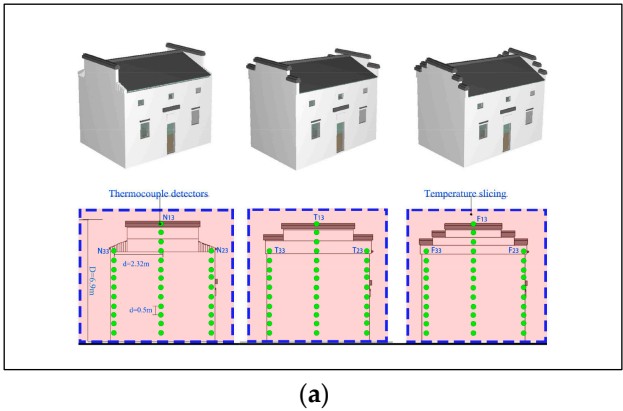 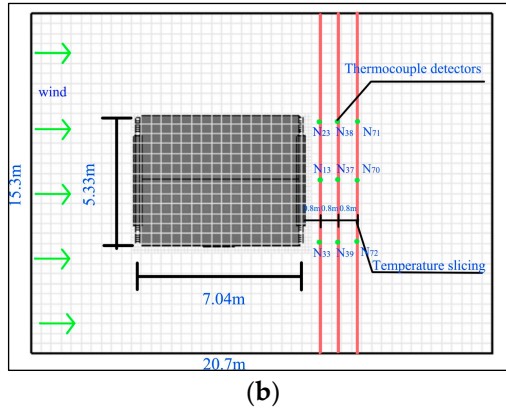

(**a**)          (**b**)

**Figure 5.** Testing equipment arrangement: (**a**) Detector arrangement; (**b**) temperature slice arrangement.

**Table 3.** Location of the detectors.

| Category | Building Spacing (m) | Highest Point in the Middle | Highest Point on the Left | Highest Point on the Right |
|---|---|---|---|---|
| fire seal wall with a single eave | 0.8 | N13 | N33 | N23 |
| | 1.6 | N37 | N39 | N38 |
| | 2.4 | N70 | N72 | N71 |
| fire seal wall with triple eaves | 0.8 | T13 | T33 | T23 |
| | 1.6 | T37 | T39 | T38 |
| | 2.4 | T70 | T72 | T71 |
| fire seal wall with quintuple eaves | 0.8 | F13 | F33 | N23 |
| | 1.6 | F37 | F39 | F38 |
| | 2.4 | F70 | F72 | F71 |

### 2.2.3. Fire Source Setting

The setting of the fire source is a crucial factor affecting the accuracy of the fire numerical simulation results, and it is reasonable to use the $t^2$ fire growth model because most traditional Huizhou buildings are wooden beam structures. The $t^2$ fire growth model can be calculated using the maximum heat release rate during a fire [28]. It can be expressed as follows:

$$Q = \alpha t^2, \tag{1}$$

where $Q$ is the heat release rate of the fire source in kW; $t$ is the time in units of s; and $\alpha$ is the fire growth coefficient in units of $kW/s^2$. According to the Chinese Technical Standard for Smoke Prevention and Exhaust Systems (GB51251-2017), the fire growth coefficient can be divided into 0.1878 $kW/s^2$, 0.04689 $kW/s^2$, 0.01172 $kW/s^2$, and 0.00293 $kW/s^2$, corresponding to superhigh, high, medium, and low speeds, respectively (Table 4). After field research, the traditional buildings in Huizhou have been weathered and eroded for hundreds of years, and the moisture content in wooden structures is low, consistent with the characteristics of fast-burning materials. Therefore, a fire growth coefficient of 0.04689 $kW/s^2$ is used in this paper, with a maximum heat release rate of 8 MW (Table 5). $t$ is calculated as 413 s, which can reach the stable burning time, and the simulation time of the whole process was set to 1200 s. The location of the fire source was set at the center

of the interior of the first floor of the traditional house, and the fire source area size is 0.5 m × 0.5 m (Figure 6).

**Table 4.** Common fire growth factors.

| Growth Type | A(kW/s$^2$) | Typical Combustible Materials |
|---|---|---|
| superhigh speed | 0.1878 | Oil pool fire, flammable decorative home |
| high speed | 0.04689 | Wooden shelf pallets, foam |
| medium speed | 0.01172 | Cotton and polyester items, wooden offices |
| low speed | 0.00293 | Heavy wood products |

**Table 5.** Maximum heat release rate values for typical fire locations.

| Typical Fire Locations | Maximum Heat Release Rate/MW |
|---|---|
| Shopping malls with sprinklers | 5.0 |
| Offices and guest rooms with sprinklers | 1.5 |
| Public places with sprinklers | 2.5 |
| Supermarkets and warehouses with sprinklers | 4.0 |
| Offices and rooms without sprinklers | 6.0 |
| Public places without sprinklers | 8.0 |
| Supermarkets and warehouses without sprinklers | 20.0 |

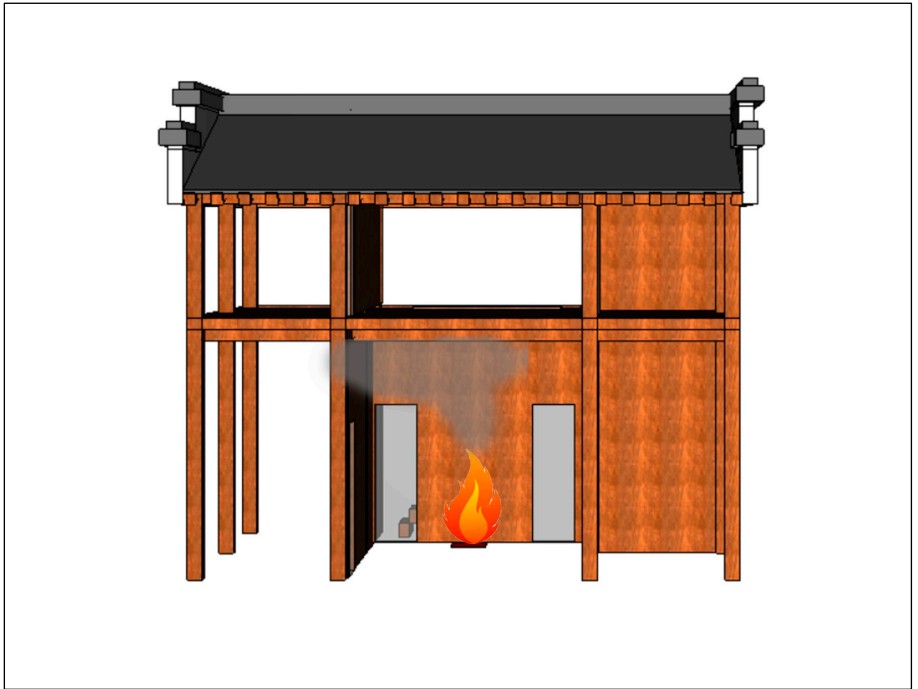

**Figure 6.** Location of the fire source.

*2.3. Grid Settings*

2.3.1. Grid Division

The grid is the smallest computational unit of the FDS tool in the process of numerical fire simulation, whose size division directly affects the reliability and accuracy of the simulation results. The smaller the grid size, the more accurate the calculation results, but this increases the calculation time and economic costs. The larger the grid size, the larger the error of the calculated results. According to Kevin [29] and others, more accurate results can be obtained when the grid size ratio to the minimum grid size $\delta x$ is 4 to 16 times. where the characteristic diameter of the fire source can be obtained as follows:

$$D^* = \left(\frac{Q}{\rho_0 c_P T_0 \sqrt{g}}\right)^{\frac{2}{5}}, \tag{2}$$

where $Q$ is the heat release rate (8000 kW), $\rho_0$ is the air density (1.206 kg/m³), $c_P$ is the specific heat capacity of air [1.005 kJ/(kg·K)], $T_0$ is the initial ambient temperature 293 K (20 °C), and $g$ is the acceleration of gravity (9.8 m/s²). The calculation indicated that $D^* \approx 1.395$ m , and the grid yields more accurate results in the interval of 0.08~0.35 m. Therefore, the 0.15 m grid selected in this paper is sufficient; however, according to the study of Zhou Qing [30] on the effect of the grid size on the FDS simulation results, the impact of the grid size on the simulation results decreases with increasing distance between the test point and the fire source. The measurement accuracy can be improved by nearly 5% when the grid size is reduced by four times. Thus, to ensure the accuracy and economy of the calculation results, the grid is divided into two suites in this paper (Figure 7), which involve the use of a finer grid of 0.15 m × 0.15 m × 0.15 m near the fire source (black areas) and a coarser grid of 0.45 m × 0.45 m × 0.45 m far from the fire source (white areas).

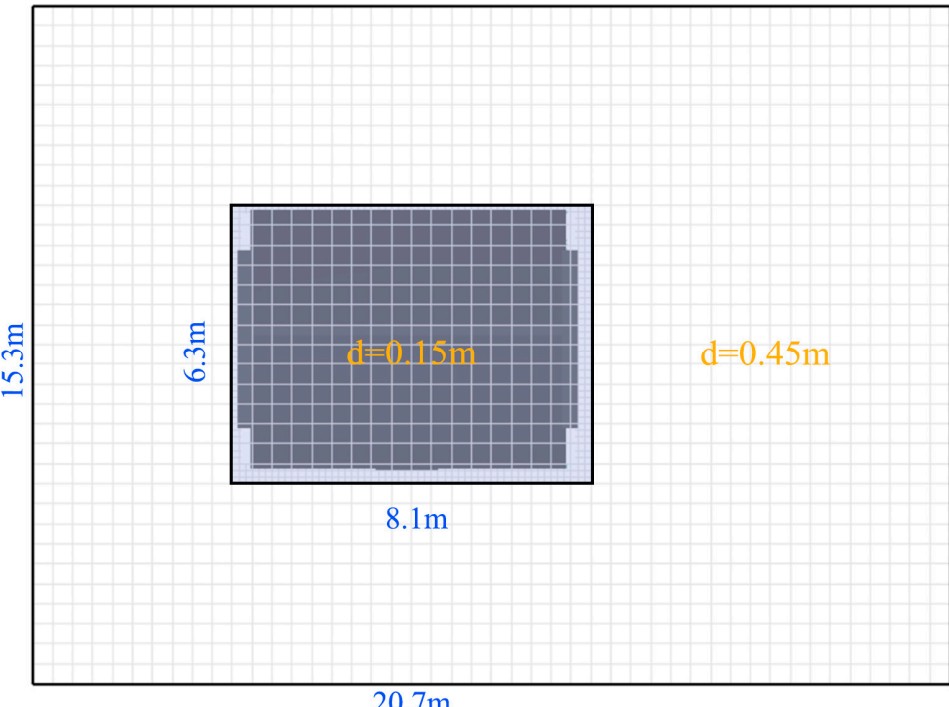

**Figure 7.** Grid division.

2.3.2. Grid Independence Verification

To ensure the accuracy and rationality of the grid division, five grid sizes (0.10 m, 0.15 m, 0.20 m, 0.30 m, and 0.45 m) were selected in this paper before the numerical simulations. The temperature detector distribution along the vertical direction at 1.8 m (groups ①), 3.6 m (groups ②), and 5.4 m (groups ③) southward of the fire source center was selected to verify the FDS grid independence (Figure 8a). The final simulation was carried out until 413 s, when the heat release rate reached its maximum value, after which the reaction stabilized. The analysis results at 450 s after the start of the simulation were selected (Figure 8b–d), which showed that the temperature distribution curve of the 0.15 m grid size increased gently. The results were stable for each group. The 0.10 m, 0.20 m, 0.30 m, and 0.45 m grid sizes showed significant volatility, among which the 0.45 m grid showed the most considerable volatility near the fire source. However, with increasing distance, the farther away from the fire source location, the smaller the curve fluctuation range and the closer the measurement results of the five grids are. Therefore, in this paper,

it is reasonable to consider the economy and accuracy of grid division and choose a 0.15 m grid near the fire source and a 0.45 m grid far from the fire source.

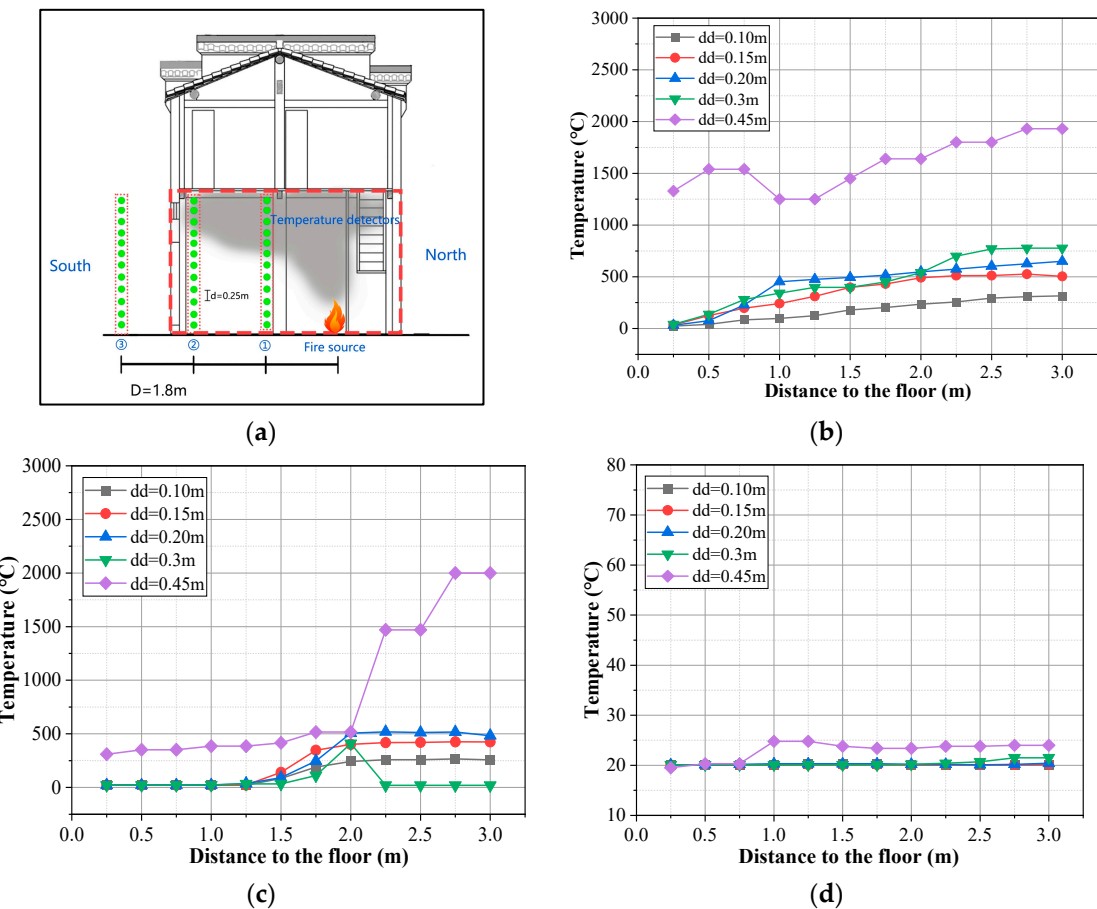

**Figure 8.** Grid independence verification. (**a**) Detector arrangement; (**b**) Group ① Simulation results; (**c**) Group ② Simulation results; (**d**) Group ③ Simulation results.

### 2.4. Simulation Verification

This work relies on experimental data for footing combustion in wooden buildings by Gao Xianzhan [31] to confirm the reliability of the FDS numerical simulation of fire in ancient structures. The same model is built with PyroSim for FDS numerical simulation to compare the FDS simulation results to the experimental results and verify the reliability of FDS in the numerical simulation of fire in ancient buildings. Gao's combustion experiment was conducted in a traditional house with a brick and wood structure in Yunnan Province, China. Burning tests were conducted in a room in the house, which is 3 m long, 4 m wide, and 2.6 m high; the ceiling is 2.2 m from the floor; and the partition walls, roof, and top of the house are all made of wood. This experiment defined the wood pile (each 0.6 m long, with a cross-sectional area of 0.025 m × 0.035 m) as the source of fire, placed vertically across to achieve steady burning. In addition, four thermocouples (S1-1, S2-2, S3-3, and S4-4) with height coordinates of 2.2 m, 1.7 m, 1.2 m, and 0.7 m, respectively, from the top to the bottom of the ceiling were positioned at the corners of the stairs along the vertical direction. The burning process lasted 8 min before the firefighters extinguished the fire.

In this paper, numerical simulations were performed using FDS under the same conditions according to the actual model size, and thermocouple detectors were arranged at the same locations, namely, THCP1, THCP2, THCP3, and THCP4, from top to bottom, respectively. (Figure 9). In the simulation, the boundaries are open, and the environmental parameters such as the wind speed, room temperature, and certain material parameters were assigned default software values. Considering the actual size of the model, the simulation

grid area is defined as 4 m × 5 m × 3 m, and the grid size is set to 0.1 m × 0.1 m × 0.1 m to ensure the accuracy of the results and the economy of the calculation process. The heat release rate of the fire source was set to 1000 kW/s$^2$, and the fire source growth model was adopted in the $t^2$ mode, in which the maximum heat release rate is reached in 350 s. Referring to Gao's experiment, the reaction from the beginning of the wood pile fire source to the end of the whole response is 600 s; in this paper, the simulation time is set to 600 s, after which the simulation is validated, and the automatic sprinkler system starts to extinguish the fire after 500 s to imitate the firefighter in Gao's experiment.

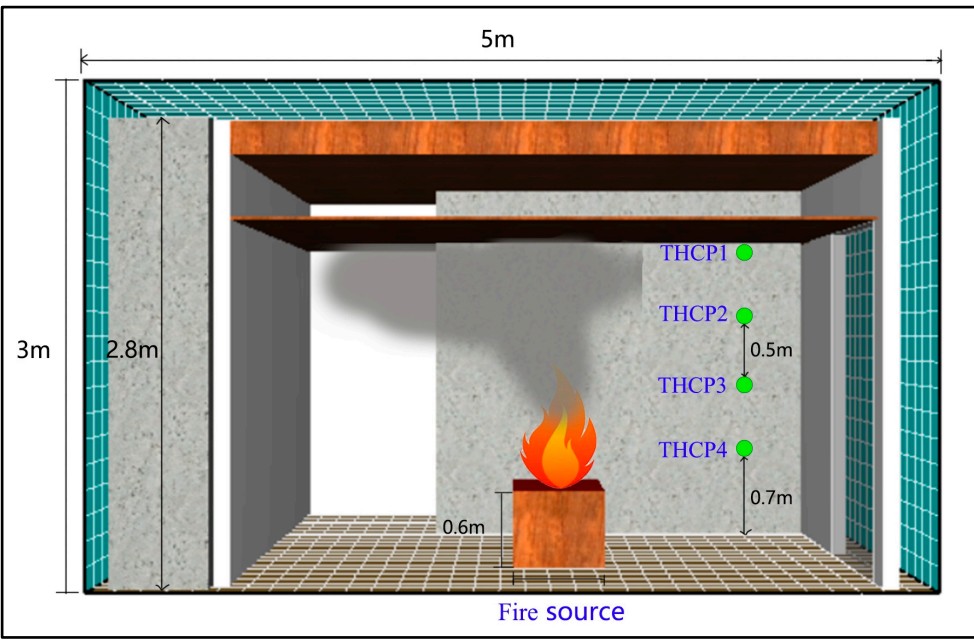

**Figure 9.** Numerical simulation scenarios.

In addition, because the different wood parameters and reference temperatures could lead to inconsistent simulation results, the study selected three experimental scenarios (Table 6) for simulation in this paper. Finally, the FDS simulation results were compared to the experimental data of Gao (Figure 10); the simulation results of Group 3 were similar to those of Group 1, the simulation results of Group 2 and the experimental results were closer, and the highest temperature in the experiment could reach 700 °C. However, overall, the simulation results are higher than the experimental results. The time to validate the data is also earlier than the experimental combustion time because the actual combustion experiment was conducted in an unsteady-state environment. The real-life indoor temperature is low, reflecting its slow occurrence over time. The external wind speed and air pressure can interfere with the experiment. At the same time, the numerical simulation in this paper was conducted under ideal steady-state conditions, which are not affected by external factors. The material parameters are default software values, which can result in errors relative to the actual values. In addition, the simplification of the model affects its accuracy. Therefore, the final numerical simulation results are significant, and the reaction occurred 600 s earlier than in Gao's experiment. However, the FDS numerical simulation results are consistent with the overall fire development trend and the maximum temperature reached, and the absolute and relative errors are within the allowable range. Therefore, it is feasible to use FDS for numerical simulation of fire in Huizhou fire seal walls, and the reference temperature of 100 °C and the reaction heat of 6000 kJ/kg selected for the wood thermal parameters in this paper are consistent with the actual values. They can be used as numerical simulation parameter conditions. The reaction time of 1200 s in the numerical simulation above can be interpreted as 1200 s after the material starts to react.

**Table 6.** Reference temperature for wood combustion.

| Group | Reference Temperature/°C | Heat of Reaction/(kJ/kg) |
|---|---|---|
| 1 | 150 | 5000 |
| 2 | 150 | 6000 |
| 3 | 100 | 6000 |

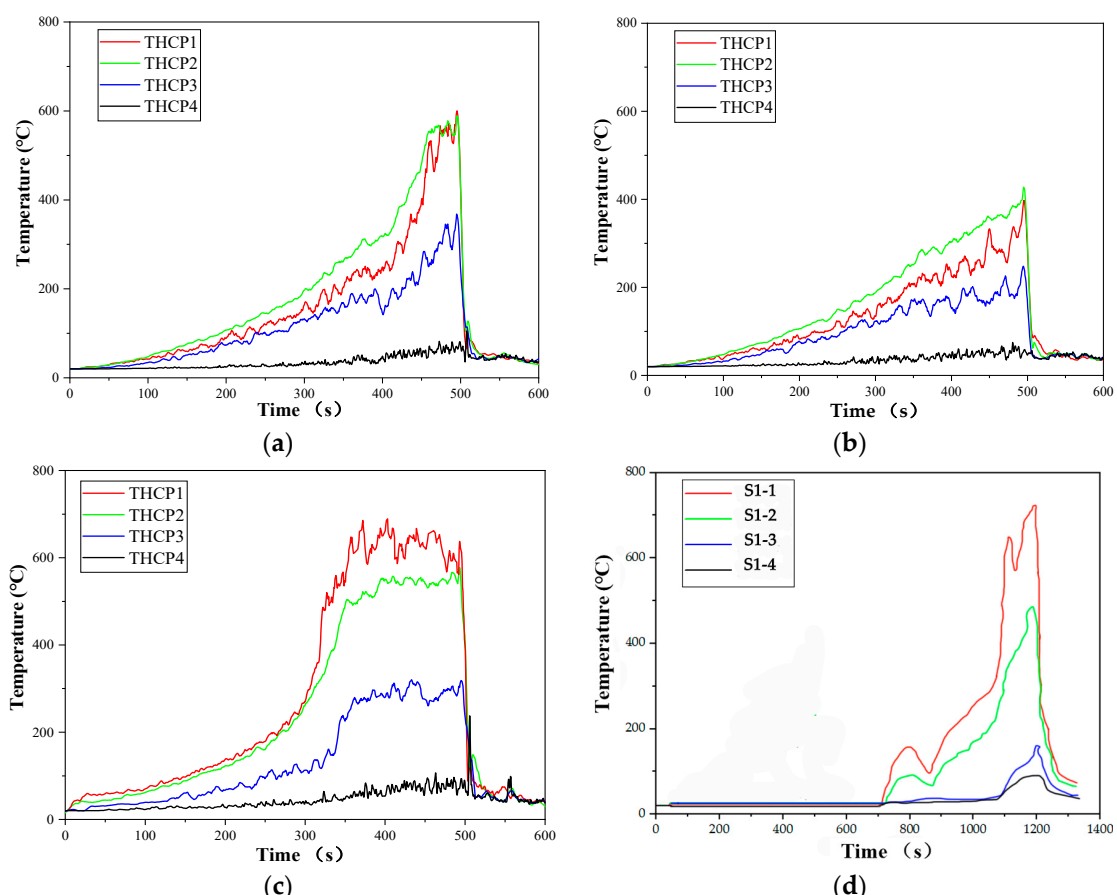

**Figure 10.** Graph of the temperature variation between the simulation and experimental results. (**a**) Group 1 temperature variation graph; (**b**) Group 2 temperature variation graph; (**c**) Group 3 temperature variation graph; (**d**) Graph of the experimental temperature change.

## 3. Results and Discussion

### 3.1. Comparative Analysis of the Fire Performance of Different Fire Seal Walls

To compare the differences in the fire performance of fire seal walls with single eaves, triple eaves, and quintuple eaves under the influence of the same environmental factors and provide a reference for the fire retrofitting of different fire seal walls, this study selected the most common scenes—F2, F5, and F8 of 27—simulation scenarios to analyze the smoke spread and temperature transfer results. In the simulation scenario, the parameters were set to be the same, including a wind speed of 1.6 m/s, a temperature of 20 °C, a building spacing of 1.6 m, and a vertical ridge height of 0 m.

#### 3.1.1. Analysis of Smoke Spread

A fire in a traditional wood building burns rapidly and produces toxic, hazardous, and high-temperature smoke. Once this smoke spreads, it can threaten human safety and damage adjacent buildings. Therefore, in this paper, smoke spreading was selected as an evaluation metric to assess the difference in fire performance between different fire seal walls.

Figure 11 shows the smoke spread of the three types of fire seal walls. In terms of combustion within 150 s, the three scenarios are similar. This may be attributed to the fact that combustion is in its initial stages and smoke diffusion is slow. From 150 to 750 s, the fire seal wall with triple eaves (scenario F5) spread smoke faster than the fire seal walls with single eaves (scenario F2) and quintuple eaves (scenario F8), and the smoke's spread could cover the entire wall. This may be attributed to the fact that the façade form of the fire seal wall with triple eaves provides weaker smoke protection than that of the fire seal wall with a single eave and quintuple eaves. From 750 to 1200 s, the smoke spread of the three kinds of fire seal walls tended to remain stable. This may be attributed to the fact that combustion is in a stabilization phase. Therefore, it could be preliminarily concluded that, under the same conditions, the type of fire seal walls has less influence on smoke protection in the early and stabilized stages of a fire. However, the smoke-blocking effect and fire performance of the fire seal wall with triple eaves are worse than those of the fire seal wall with single and quintuple eaves during the fire growth phase.

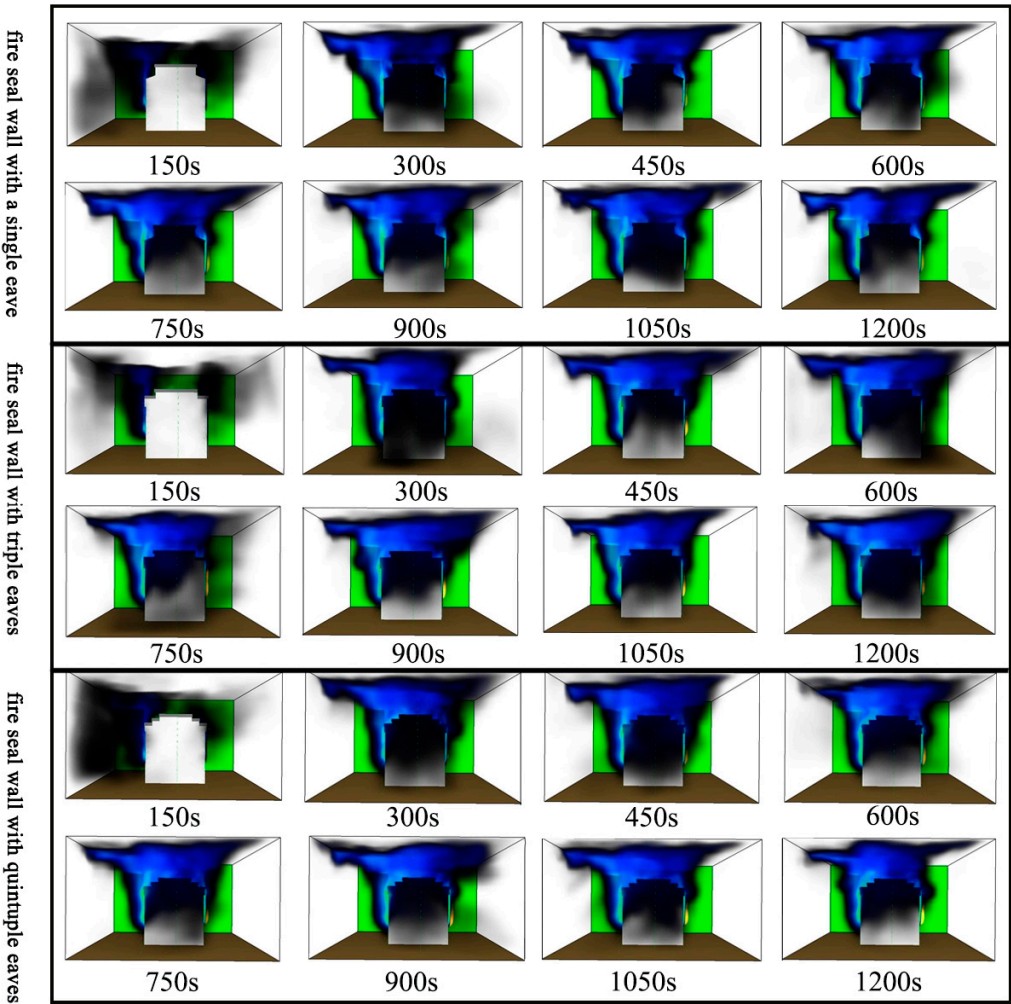

**Figure 11.** Smoke spread of the three types of fire seal walls.

3.1.2. Analysis of Temperature Transfer

A traditional wooden building produces substantial heat in the burning process, transferring thermal radiation to the surrounding area. The temperature of the adjacent wooden building reaches 260 °C, which easily produces a fire. Therefore, this study analyzed the heat-blocking effect of different fire seal walls to determine the difference in fire performance between the various types of Huizhou fire seal walls.

Figure 12a shows the results of the average temperature change curve of adjacent measurement points. From 0 to 400 s, the temperature increase rate of the fire seal wall with triple eaves is faster than that of the other two walls. From 400 to 1200 s, the average temperature of all three kinds of fire seal walls steadily fluctuates above and below 32 °C, which is well below the flash ignition temperature of wooden buildings. This finding aligns with previous research that reports the fire performance of fire seal walls with triple eaves [32]. In addition, the average temperature of the fire seal wall with a single eave is lower than that of the other two walls. This may be attributed to the fact that fire seal walls have good fire resistance, and the fire resistance of the fire seal wall with a single eave is higher than that of the other two walls. In addition, as shown in the temperature change cloud (Figure 13), the heat transferred by the fire is mainly concentrated at the top of the fire seal wall, and more heat is concentrated at the highest point on the left side. This may be attributed to the fact that fire resistance is weaker at the top of the fire seal wall compared to other parts of the wall.

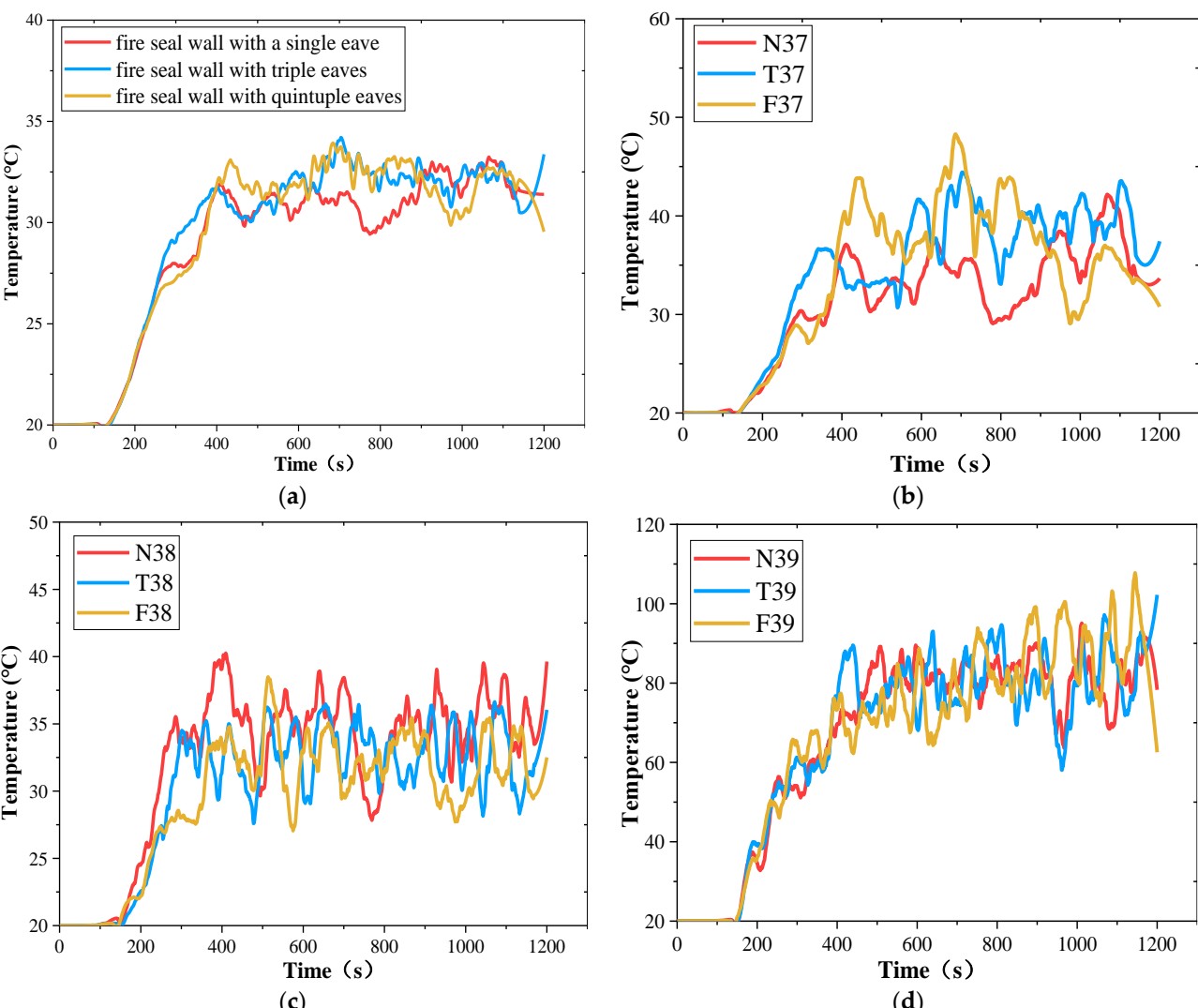

**Figure 12.** Temperature change curve of the detector at the different positions. (**a**) The average temperature change curve of adjacent measurement points; (**b**) The temperature change curve of the detector at the highest point in the middle; (**c**) Temperature change curve of the detector at the highest point on the right; (**d**) Temperature change curve of the detector at the highest point on the left.

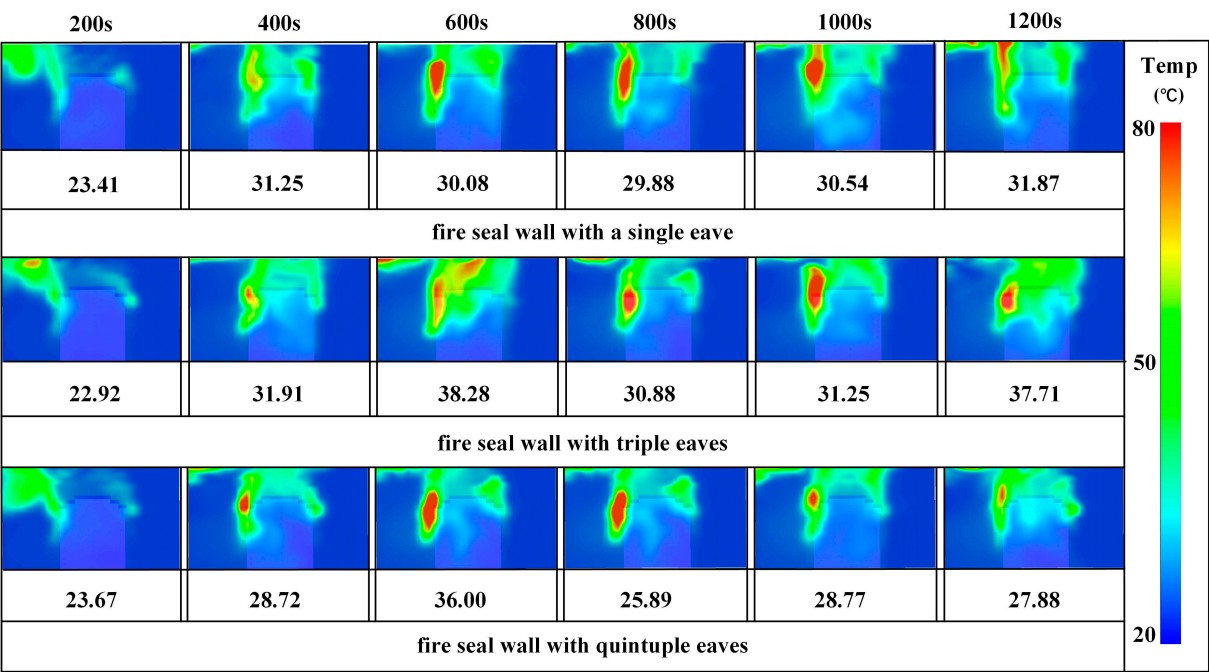

**Figure 13.** Temperature variation clouds of adjacent slices.

To further study the difference in fire performance between the tops of different fire seal walls under the same conditions, the detector at the highest point in the middle of the fire seal wall, the highest point on the left, and the highest point on the right were selected for in-depth analyses (Figure 12b–d). With respect to the temperature change trend, from 0 to 200 s, the curves for the three kinds of fire seal walls are close. From 200 to 1200 s, the temperature at the highest point on the left side of the fire seal wall is higher than that at the highest point in the middle and the highest point on the right side. In addition, the temperature change trend of the maximum middle moment of the fire seal wall with a single eave is slightly lower than those of the fire seal walls with triple and quintuple eaves; the temperature change trend of the maximum right point is higher than that of the fire seal walls with triple and quintuple eaves; and the temperature change trend of the left complete end is the same for all three wall types.

To further quantify the rapidity of fire spread at different locations on top of the fire seal walls, this study defines the ratio of the maximum temperature achieved by the detector to the time taken to reach the peak as the average temperature increase rate. As shown in Figure 14 and Table 7, the average temperature increase rate of the fire seal wall with triple eaves is higher than that of the fire seal walls with quintuple eaves, and the fire seal walls with quintuple eaves are higher than that of the fire seal wall with a single eave. In addition, the average temperature increase rate at the highest point on the left side of the three fire seal walls is higher than that at the highest point in the middle and the highest point on the right side. This may be attributed to the fact that the greater area of windows and doors on the south side of the building results in quicker conduction of heat, resulting in a more rapid increase in temperature at the highest point on the left side of the fire seal wall. Thus, it could be concluded that, under the same conditions, the performance of the fire seal wall with single eaves is superior to that of the fire seal walls with quintuple eaves in terms of performance, and the fire seal walls with quintuple eaves are superior to fire seal walls with triple eaves in the middle and late stages of a fire. Furthermore, fire performance at the highest point on the left side of the three fire seal walls is weaker than that at the highest point in the middle and on the right side.

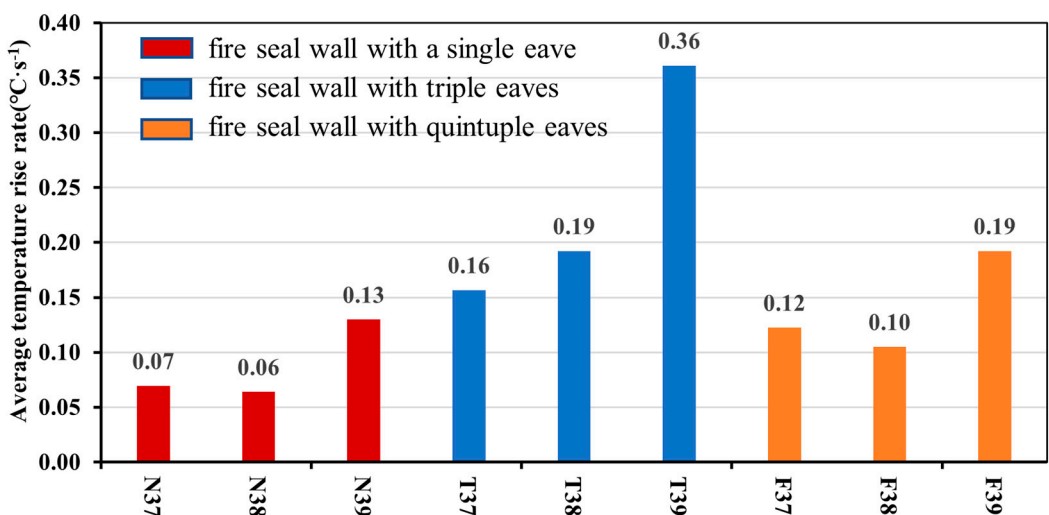

**Figure 14.** Average temperature versus the rate of detection at the different locations.

**Table 7.** Temperature variation of the detectors on the different fire seal walls under the same conditions.

| Schematic | Scenes | Detector | Peak point Temperature (°C) | Time to the Peak (s) | Average Temperature Rise Rate (°C·s$^{-1}$) |
|---|---|---|---|---|---|
| | | N37 | 74.27 | 1070.41 | 0.07 |
| | | N38 | 59.81 | 936.01 | 0.06 |
| | F2 | N39 | 128.72 | 992.40 | 0.13 |
| | | T37 | 93.21 | 595.20 | 0.16 |
| | | T38 | 54.65 | 284.40 | 0.19 |
| | F5 | T39 | 154.70 | 428.41 | 0.36 |
| | | F37 | 83.22 | 680.41 | 0.12 |
| | | F38 | 55.00 | 524.41 | 0.10 |
| | F8 | F39 | 155.61 | 810.01 | 0.19 |

### 3.2. Analysis of the Impacts of Different Factors on the Fire Performance of Fire Seal Walls

From the above, under the same conditions, the overall and top fire performance of the fire seal wall with single eaves is superior to that of fire seal walls with quintuple eaves, and fire seal walls with quintuple eaves are superior to those with triple eaves in the middle and late stages of a fire. To further study the fire performance of fire seal walls, this study selected a fire seal wall with a single eave, which has the best fire performance, and a fire seal wall with triple eaves, which has the weakest fire performance, in order to analyze the effects of different wind speeds (Scenarios W1–W3) on the fire performance of fire seal walls with single eaves; moreover, the impacts of different building spacings (Scenarios F4–F6) and vertical ridge heights (Scenarios H4–H6) on the fire performance of fire seal walls with triple eaves were also analyzed.

### 3.2.1. Analysis of the Effect of Different Wind Speeds on Smoke Spread

Figure 15 shows the smoke spread of the fire seal wall with a single eave under different wind speed scenarios. In scenario W1, under windless conditions (Figure 15a), the smoke spreads uniformly around the building, and the spread is greater on the southern side. This may be attributed to the fact that the lack of wind in the area causes fires to spread slower in the building. Furthermore, more doors and windows are located on the southern

side of the building than on the northern side. Most indoor heat is transferred outward via doors and windows on the southern side, which causes a more favorable range of temperature changes on the southern side of the building. In scenario W2, under the wind speed condition of 1.6 m/s (Figure 15b), the wind accelerated the fire spread. Substantial amounts of smoke spread toward the right side of the fire seal wall, and most of the heat transferred by the fire was concentrated above the fire seal wall. In scenario W3, under the 7 m/s wind speed condition (Figure 15c), the smoke diffusion toward the northern and southern sides of the building increases, and the amount of spread toward the right side of the building is significantly reduced. This behavior is consistent with experiments on the effect of wind speed on fire spread [33]. This may be attributed to the fact that the stronger wind blows away a large amount of heat and smoke, which slows the spread of fire to the right side of the building. Therefore, it can be concluded that the windless scenario and the higher wind speed scenario have less impact on the fire performance of the fire seal wall. Moreover, the lower wind speed scenario has a greater impact on the fire performance of the fire seal wall, which can promote the spread of fire to neighboring buildings.

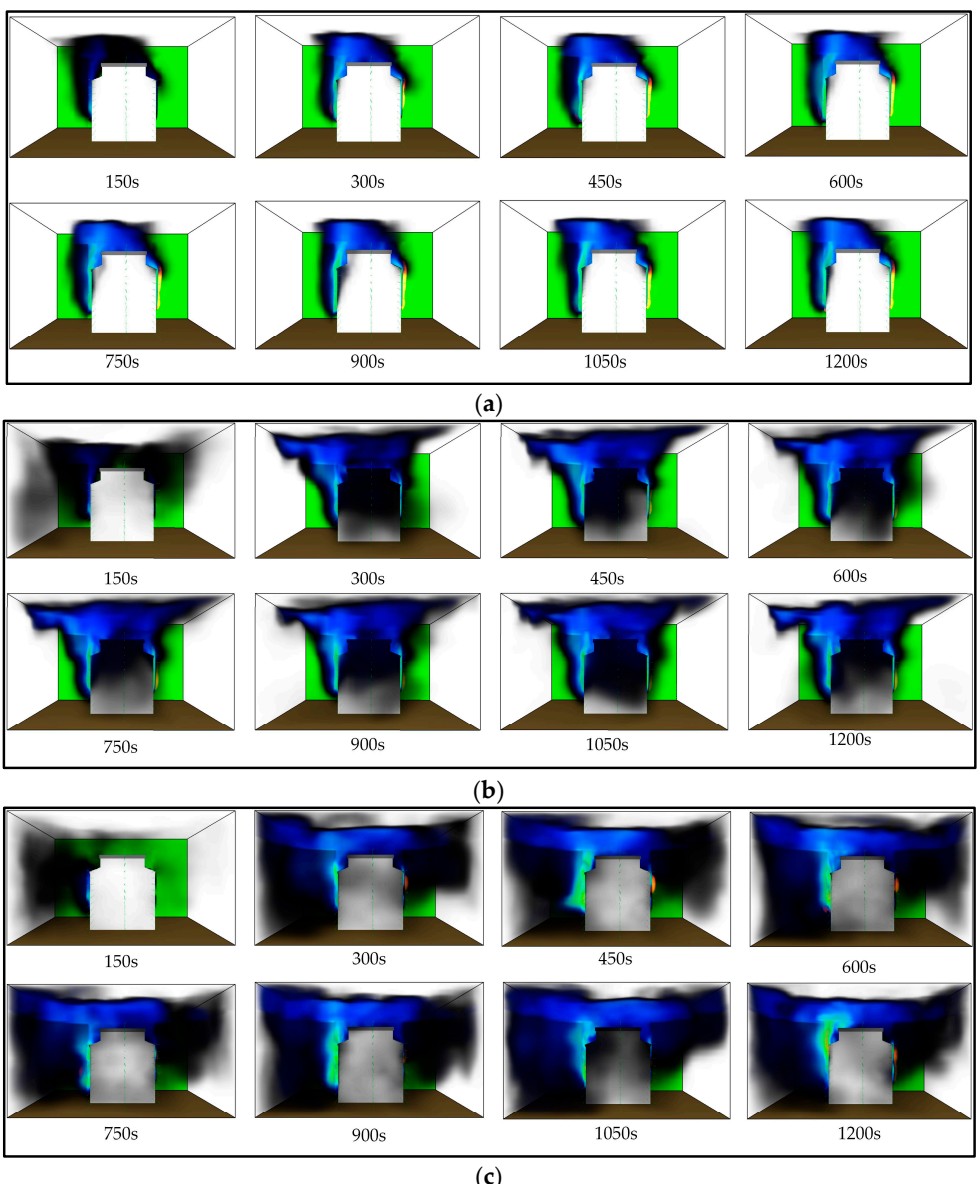

**Figure 15.** Smoke spread of the fire seal wall with a single eave under the different wind speed scenarios. (**a**) Smoke spread under the 0 m/s wind speed scenario; (**b**) Smoke spread under the 1.6 m/s wind speed scenario; (**c**) Smoke spread under the 7 m/s wind speed scenario.

### 3.2.2. Analysis of the Effect of the Different Building Spacings and Vertical Ridge Heights on Temperature Transfer

Traditional Huizhou villages are densely laid out with different alley widths; the spacing is usually 0.8 m, 1.6 m, and 2.4 m, and there are different types of fire seal walls surrounding the buildings with varying ridge heights, e.g., usually 0 m, 0.5 m, and 1 m, which are the key factors affecting the spread of fire in traditional Huizhou villages and buildings. Therefore, we selected the building spacing and vertical ridge height as the key factors influencing the fire seal wall's performance.

Figure 16 and Table 8 show the average temperature variation of the fire seal wall with triple eaves under different building spacings and vertical ridge heights. In scenarios F4–F6, from 0 to 200 s, the simulation results are similar for three different building spacings. From 200 to 1200 s, the average temperature of the detectors with respect to the 2.4 m building spacing is lower than the 0.8 m and 1.6 m scenarios. In scenario H4–H6, from 0 to 800 s, the average temperature of the detector decreases with increasing vertical ridge height. From 800 to 1200 s, the average detector temperature remains stable. This finding aligns with previous research that reports a decrease in temperature with respect to increasing vertical ridge height and building spacing [34].

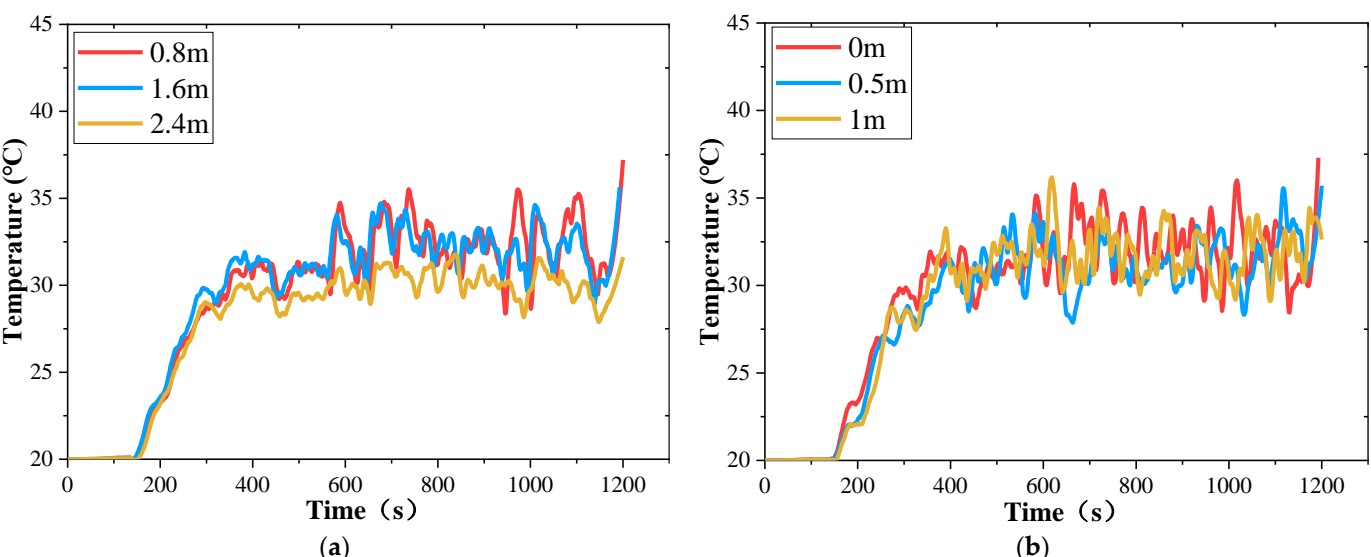

**Figure 16.** Temperature change curve of the fire seal wall with triple eaves under the different factor scenarios. (**a**) Temperature change curves under the different building spacing scenarios; (**b**) Temperature variation curves under the various vertical ridge height scenarios.

**Table 8.** Average temperature variation of the fire seal wall with triple eaves under the different building spacings and vertical ridge heights.

| Scenes | Time | | 200 s | 400 s | 600 s | 800 s | 1000 s | 1200 s |
|---|---|---|---|---|---|---|---|---|
| F4 | | 0.8 m | 22.84 | 32.90 | 31.98 | 32.82 | 29.54 | 35.88 |
| F5 | Building spacing | 1.6 m | 22.92 | 31.91 | 38.28 | 30.88 | 31.25 | 37.71 |
| F6 | | 2.4 m | 22.90 | 30.27 | 30.47 | 26.65 | 30.09 | 28.99 |
| H4 | | 0 m | 22.92 | 31.91 | 38.28 | 30.88 | 31.25 | 37.71 |
| H5 | Vertical ridge height | 0.5 m | 22.38 | 32.51 | 29.53 | 30.34 | 28.93 | 34.33 |
| H6 | | 1 m | 22.36 | 31.35 | 28.74 | 26.64 | 29.83 | 35.47 |

Figure 17 and Table 9 show the average temperature increase rate of the different locations on top of the fire seal wall with triple eaves with different building spacings and vertical ridge heights. In scenarios F4–F6, the average temperature increase rate of the top of the fire seal wall with triple eaves with respect to the 1.6 m building spacing was

higher than that under the 0.8 m and 2.4 m scenarios. In addition, with increasing building spacing, the average temperature increase rate at both the middle of the fire seal wall with triple eaves and at the highest point on the left side of the screen wall showed an increasing and then decreasing trend, and the average warming rate at the highest point on the right side gradually increased. In scenarios H4–H6, the average temperature increase rate of the top of the fire seal wall with triple eaves with respect to 0 m vertical ridge heights was higher than that of the 0.5 m and 1m scenarios. In addition, with the increasing height of the vertical ridge, the average temperature increase rate at the highest point in the middle and the highest point on the right side of the fire seal wall with triple eaves showed a decreasing and then increasing trend. Then, the increasing trend and average warming rate at the highest point on the left side showed apparent decreasing trends.

**Table 9.** Temperature change of the fire seal wall with triple eaves under the different factor scenarios.

| Schematic | Scenen | Detector | Peak Point Temperature (°C) | Time to the Peak (s) | Average Temperature Rise Rate (°C·s$^{-1}$) |
|---|---|---|---|---|---|
|  | F4 | T13 | 90.72 | 975.6 | 0.09 |
| | | T23 | 56.45 | 1105.2 | 0.05 |
| | | T33 | 210.63 | 1196.4 | 0.18 |
|  | F5 | T37 | 93.21 | 595.2 | 0.16 |
| | | T38 | 54.65 | 284.4 | 0.19 |
| | | T39 | 154.7 | 428.41 | 0.36 |
|  | F6 | T70 | 77.4 | 681.6 | 0.11 |
| | | T71 | 53.78 | 284.4 | 0.19 |
| | | T72 | 103.51 | 1015.2 | 0.1 |
|  | H4 | T37 | 93.21 | 595.2 | 0.16 |
| | | T38 | 54.65 | 284.4 | 0.19 |
| | | T39 | 154.7 | 428.41 | 0.36 |
|  | H5 | T37 | 76.17 | 1149.61 | 0.07 |
| | | T38 | 61.98 | 1118.4 | 0.06 |
| | | T39 | 138.87 | 984.01 | 0.14 |
|  | H6 | T37 | 88.84 | 728.41 | 0.12 |
| | | T38 | 55.89 | 614.41 | 0.09 |
| | | T39 | 124.07 | 877.21 | 0.14 |

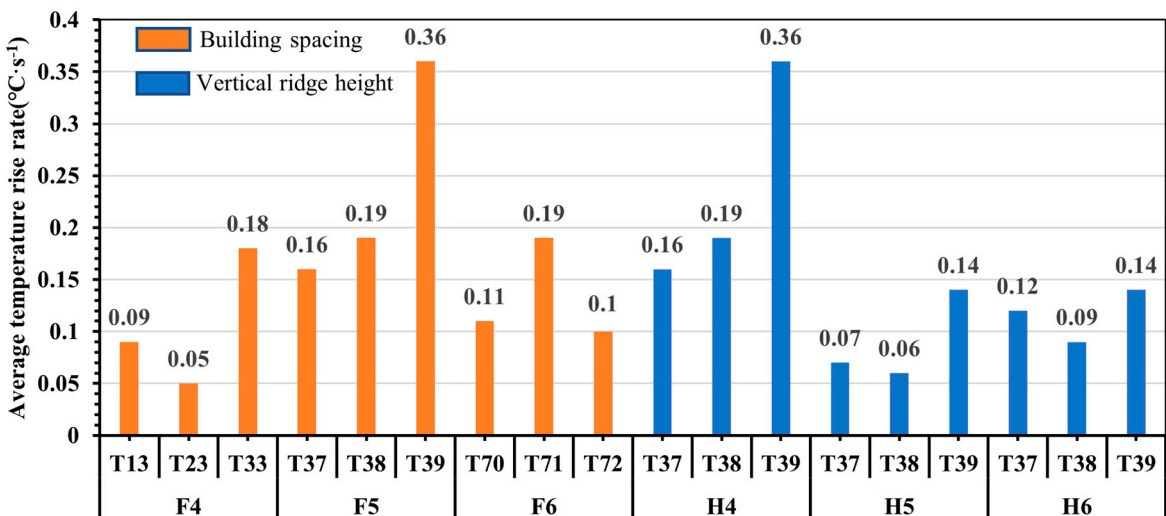

**Figure 17.** The average temperature rise rate of the top of the fire seal wall with triple eaves at different building spacings and vertical ridge heights.

Therefore, it could be concluded that, with increasing building spacing and vertical ridge height, the fire seal wall provided better overall fire performance during the fire's growth and stabilization phase. In addition, the highest point in the middle, the highest point on the left, and the highest point on the right of the fire seal wall had the worst fire performance with respect to the 1.6 m building spacing and 0 m vertical ridge height scenarios.

## 4. Conclusions

Twenty-seven fire simulation scenarios under the influence of different wind speeds, building spacings, and vertical ridge heights were constructed to analyze the characteristics of the fire performance of different types of Huizhou fire seal walls. The simulation's conclusions are as follows:

1.  The results of the fire simulation scenarios at different fire seal walls show that, under the same conditions, the fire seal wall with single eaves is superior to that of the fire seal walls with quintuple eaves in terms of performance, and the fire seal walls with quintuple eaves are superior to the fire seal wall with triple eaves in the middle and late stages of a fire. Furthermore, the fire performance at the highest point on the left side of the three fire seal walls is weaker than that at the highest point in the middle and on the right side.
2.  The results of fire simulation scenarios at different wind speeds show that the windless scenario and the higher wind speed scenario have less impact on the fire performance of the fire seal wall. Moreover, the lower wind speed scenario has a greater impact on the fire performance of the fire seal wall, which can promote the spread of fire to neighboring buildings.
3.  The results of the fire simulation scenarios at different building spacings and vertical ridge heights show that, with increasing building spacing and vertical ridge height, overall, the fire seal wall provided better fire performance during the fire increase and stabilization phase. In addition, the highest points in the middle and on the left and right of the fire seal wall had the worst fire performance with respect to the 1.6 m building spacing and 0 m vertical ridge height scenarios.
4.  When performing the fire retrofitting of emblematic fire seal walls, the focus should be on repairing the top of the fire seal wall with triple eaves and near areas with many windows and doors. When performing the fire retrofitting of emblematic fire seal walls for areas with low wind speeds year-round, the focus should be on fire protection between neighboring buildings.

5. An increase in the height of the vertical ridge and the fire separation distance can be used as a fire protection retrofitting measure for future fire seal walls. Moreover, for fire seal walls with a vertical ridge height of 0 m and a fire separation distance of 1.6 m, measures such as brushing fireproof paint can be adopted to strengthen the fire protection performance of the weak points at the top of the wall.

This study contributes to improving the fire protection renovation of Huizhou fire seal walls. However, this study has certain limitations. For example, the Huizhou fire seal wall in the traditional fire protection system was simplified into three types, namely, fire seal walls with single eaves, triple eaves, and quintuple eaves, and the model comprised a simplified primary wall type. In addition, only 27 simulated scenarios were constructed for analysis, and it is impossible to completely simulate and analyze fire seal walls considering all styles and fire scenarios. Therefore, this study only provides reference ideas for optimizing and transforming traditional fire protection technology.

**Author Contributions:** Conceptualization, Y.W. and B.H.; methodology, Y.W. and B.H.; software, B.H.; formal analysis, B.H. and S.C.; investigation, Y.W., S.C. and J.Y.; data curation, B.H., S.C. and J.Y.; writing—original draft preparation, Y.W., B.H. and S.C.; writing—review and editing, Y.W. and S.C.; visualization, Y.W. All authors have read and agreed to the published version of the manuscript.

**Funding:** This research was funded by the Natural Science Research Program of Anhui Colleges, grant number KJ2021ZD0067, the Open Project Program of the China-Portugal Joint Laboratory of Cultural Heritage Conservation Science (No. SDYY2102).

**Institutional Review Board Statement:** Not applicable.

**Informed Consent Statement:** Not applicable.

**Data Availability Statement:** The data presented in this study are available on request from the corresponding author.

**Conflicts of Interest:** The authors declare no conflict of interest.

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
