# Peer review of "FDS-Based Study of the Fire Performance of Huizhou Fire Seal Walls in Traditional Residential Buildings in Southern China"

_fire, doi:10.3390/fire6100388_

Round 1
Reviewer 1 Report
The paper presents a FDS-Based research of the fire performance of Huizhou fire seal walls. The subject is very interesting and covers a lack in the literature as not many papers deal with this typology of fire seal walls under fire. Research results in this paper can provide useful reference for improving the fire safety of traditional buildings. I recommend the paper to be accepted for publication. The following comments may be considered by the authors in revising the paper.
1. In table 4, the unit of fire growth factor should be given.
2. The conclusion can be more concise.
Reviewer 2 Report
The authors presented the results of research of the fire seal walls in traditional residential buildings in Southern China
The manuscript sent for evaluation consists of 24 pages, including 20 pages of text with 9 tables and 17 figures, and 1 pages for references. In my opinion, this article is huge and it may be selected to 2 parts.
The topic is original and relevant in the field of fire protection. the article is very rich in content.
Abstract would be to contain all the necessary information, such as introduction, aim, methods, results, main statements. Please, add aim of article.
The visual documentation is illustrative and clear.
After reading the text, in general, I assess the scientific quality of the publication as good
Methodology is very complicated, but interesting. Please, specify:
Line 366: this study selected the most common scenes F2, F5, and F8 of simulation scenarios...
Where did the said scenarios come from? Which part of the methodology does this follow?
The Results and Discussion chapter describes the obtained experimental data.
I ask the authors to supplement the discussion section by comparing the results with other authors.
The conclusions are brief. Please clearly specify the result obtained.
Line 283 are the groups labeled correctly?
I wish you every success in your future work
Reviewer 3 Report
1. The Introduction section appears to lack vibrancy, as it lacks subheadings that would aid in comprehending the study effectively. Hence, add subheadings accordingly
2. Consider including a concise summary of the study at the conclusion of the Introduction section to improve readability.
3. The quality of the figures as a whole is excellent. However, Figure 4 might require some improvements.
4. The article contains recurring errors in the English language. It is recommended to thoroughly review and revise for greater clarity and accuracy.
5. With three extended points, the Conclusion section seems to be lengthy. For greater clarity, it is suggested that the conclusion contain five to six concise points.
English language editing is required
